# HS-SPME/GC×GC-TOFMS-Based Flavoromics and Antimicrobial Properties of the Aroma Components of *Zanthoxylum motuoense*

**DOI:** 10.3390/foods12112225

**Published:** 2023-05-31

**Authors:** Wei Gu, Yinghuan Wei, Xianjie Fu, Ronghui Gu, Junlei Chen, Junyou Jian, Liejun Huang, Chunmao Yuan, Wenling Guan, Xiaojiang Hao

**Affiliations:** 1State Key Laboratory of Functions and Applications of Medicinal Plants, Guizhou Medical University, Guiyang 550014, China; we_ynhn@163.com (Y.W.); fuxjgzcdc1997@163.com (X.F.); stefanpin@163.com (J.C.); jyjian2418@163.com (J.J.); huangliejun@126.com (L.H.); yuanchunmao01@126.com (C.Y.); haoxj@mail.kib.ac.cn (X.H.); 2Natural Products Research Center of Guizhou Province, Guiyang 550014, China; 3School of Liquor and Food Engineering, Guizhou University, Guiyang 550025, China; rhgu@gzu.edu.cn; 4College of Horticulture and Landscape, Yunnan Agricultural University, Kunming 650204, China; 5State Key Laboratory of Phytochemistry and Plant Resources in West China, Kunming Institute of Botany, Chinese Academy of Sciences, Kunming 650201, China

**Keywords:** *Zanthoxylum motuoense*, volatile components, HS-SPME/GC×GC-TOFMS, flavoromics strategy, antimicrobial activities

## Abstract

*Zanthoxylum motuoense* Huang, native to Tibet, China, is a newly discovered Chinese prickly ash, which, recently, has increasingly attracted the attention of researchers. In order to understand its volatile oil compositions and flavor characteristics, and to explore the flavor difference between *Z. motuoense* and the common Chinese prickly ash sold in the market, we analyzed the essential oils of *Z. motuoense* pericarp (MEO) using HS-SPME/GC×GC-TOFMS coupled with multivariate data and flavoromics analyses. The common commercial Chinese prickly ash in Asia, *Zanthoxylum bungeanum* (BEO), was used as a reference. A total of 212 aroma compounds from the 2 species were identified, among which alcohols, terpenoids, esters, aldehydes, and ketones were the major compounds. The predominant components detected from MEO were citronellal, (+)-citronellal, and β-phellandrene. Six components—citronellal, (E,Z)-3,6-nonadien-1-ol, allyl methallyl ether, isopulegol, 3,7-dimethyl-6-octen-1-ol acetate, and 3,7-dimethyl-(R)-6-octen-1-ol—could be used as the potential biomarkers of MEO. The flavoromics analysis showed that MEO and BEO were significantly different in aroma note types. Furthermore, the content differences of several numb taste components in two kinds of prickly ash were quantitatively analyzed using RP-HPLC. The antimicrobial activities of MEO and BEO against four bacterial strains and nine plant pathogenic fungi were determined in vitro. The results indicated that MEO had significantly higher inhibitory activities against most microbial strains than BEO. This study has revealed the fundamental data in respect of the volatile compound properties and antimicrobial activity of *Z. motuoense*, offering basic information on valuable natural sources that can be utilized in the condiment, perfume, and antimicrobial sectors.

## 1. Introduction

As an important condiment, the Chinese prickly ash (Hua jiao) occupies an indispensable position in the field of food seasoning worldwide, particularly in East Asia. Hua jiao belongs to the genus *Zanthoxylum* of Rutaceae. The genus *Zanthoxylum*, which is mainly distributed in tropical to subtropical areas, comprises approximately 250 species of perennial trees or shrubs [1,2]. Of these, there are 45 species and 13 varieties in China [3,4]. The fruits of some *Zanthoxylum* species are commonly used as condiments because of their distinctive aroma, such as fresh, floral, spicy, or green, and their distinctive tingling taste [5]. In addition, the fruits of *Zanthoxylum* spp. are also commonly used in different traditional systems of medicine and in several other applications, e.g., as chemopreventive agents, for tooth care, or as perfumes [6].

However, only a few *Zanthoxylum* fruits are edible, as many *Zanthoxylum* species’ fruits contain hepatotoxic components. Therefore, it is valuable to find new resources of *Zanthoxylum* plants that can be safely consumed and have a special flavor. *Zanthoxylum motuoense* Huang is a newly discovered *Zanthoxylum* plant that has been used as a condiment source in recent years. It has distinctive characteristics owing to its unique aroma and habitat in the Qinghai-Tibet Plateau, China. *Z. motuoense* is a deciduous tree that can grow up to a height of 15 m. It is endemic to Motuo County, Tibet, China [7]. The edible and medicinal values of *Z. motuoense* were first reported in 2020 [8]. Its pericarp has a strong lemon-like aroma and warm peppery flavor, which are very different from those of common Chinese pepper. In addition, *Z. motuoense* fruit is often used as a herbal medicine to treat stomach pain, toothache, and food poisoning or to kill parasites [8]. Recently, research has been carried out on the ethanol extract components of ten typical *Zanthoxylum* herbs, including *Z. motuoense*, with the employment of UHPLC-QTOF-MS/MS [9]. However, there are still no reports on the aroma component analysis, particularly regarding the flavor substances and biological activity of *Z. motuoense* pericarp. Therefore, the research on the aroma substances and medicinal efficacy of *Z. motuoense* will be of great significance to the development of *Z. motuoense* as a new resource for edible spices and agricultural natural antimicrobial agents in the future.

Previous research on volatile compounds from *Zanthoxylum* essential oils was mainly performed by matching mass spectral data obtained via gas chromatography-mass spectrometry (GC-MS) analysis with existing databases [6]. However, 1D GC has some limitations, as co-elution frequently occurs when the compounds exhibit a similar affinity to the stationary phase, particularly for samples that contain complicated volatile fractions, and there may be more compounds with a similar affinity. This phenomenon significantly affects the accuracy of identification and quantification. Gas chromatography×gas chromatography (GC×GC) is a powerful tool for separating and analyzing a diverse range of complex samples and adds a second dimension of chromatographic resolution by using two distinct stationary phase capillary columns for a single analysis [10,11]. Consequently, GC×GC can separate considerably more compounds than conventional GC methods [12].

Flavoromics is an “omic” and “holistic” approach focused on low-molecular-mass compounds (volatile and nonvolatile) and linking them to a defined sensorial perception [13]. The flavoromics approach provides a new perspective for correlating the particular sensory attributes (odor properties) of food with its chemical composition [14].

In this study, HS-SPME and GC×GC-TOFMS coupled with a flavoromics approach were used to analyze the volatile compounds of *Z. motuoense* pericarp (MEO) and the common commercial Chinese pepper *Z. bungeanum* pericarp (BEO). In addition, the content of several typical numb-taste components in two species of *Zanthoxylum* was quantitatively compared via the HPLC method. Next, inspired by its traditional usage in the local area, the antimicrobial activities of MEO and BEO against several bacterial strains and agricultural pathogenic fungi strains were screened. Our work provides valuable data for assessing the edible and medicinal value of *Z. motuoense*, and for guiding the direction of “huajiao” industry development on the Qinghai-Tibet Plateau.

## 2. Materials and Methods

### 2.1. Zanthoxylum Samples

The pericarp of *Z. motuoense* was collected in the period October 2020 to September 2021 in Motuo, Tibet, and identified by Professor Wen-ling Guan. The pericarp of *Z. bungeanum* used in this study was collected from Mao County, Sichuan Province, in July 2021 and identified by Professor Wei Gu (Figure 1). The specimens (HZA-2021-10-3 and HZA-2021-7-8) were stored at the Key Laboratory of Chemistry for Natural Products of Guizhou and the Chinese Academy of Sciences. The pericarp was then crushed using a high-speed disintegrator (Wenling Auari Traditional Chinese Medicine Machinery Co., Ltd., Wenling, China). To prevent the loss of highly volatile compounds after sample preparation, the sample powders were prepared before analysis, sealed, and stored in an ultralow-temperature refrigerator.

### 2.2. Volatile Compounds Analysis

#### 2.2.1. HS-SPME Conditions

The samples (0.5 g each) were placed in a headspace bottle and were extracted using SPME with a 1 cm DVB/CAR/PDMS fiber head and kept in a 60 °C bath for 40 min. The extracted samples were desorbed in the GC inlet for 5 min and subjected to GC×GC-TOFMS analysis according to the set parameters.

#### 2.2.2. GC×GC-TOFMS Conditions under SPME Method

First-dimensional column: DB-WAX (30 m × 250 μm × 0.25 μm); the injection temperature was 250 °C and the initial temperature was 40 °C, which was maintained for 3 min and then increased to 250 °C at 5 °C/min and maintained for 5 min. Helium (99.9999%) 1.0 mL/min; splitless injection. Second-dimensional column: DB-17MS (2 m × 100 μm × 0.10 μm); the column temperature was always 5 °C higher than that of the first-dimension column. The modem temperature was always 5 °C higher than the first-dimension column. The modulation period was 6.0 s in full two-dimensional analysis; interface temperature was 270 °C; ion source temperature was 250 °C; electron bombardment source was 70 eV; detector was 1680 v; and collection rate was 50 sheets/s. The scanning range of mass spectrometry was *m*/*z* 33~500; NIST Spectra Database.

#### 2.2.3. Statistical Analysis

Each group of samples was tested in quadruplicate to ensure the reliability of the experimental results. GC×GC-TOFMS data analysis was performed at the Suzhou Bionovogene (Suzhou, China). The original GC×GC-TOFMS data were pre-treated using the R software v 3.3.2 platform. The edited data matrix was then imported into the ropls software packages in the R software platform for multivariate analysis, including principal component analysis (PCA), partial least-squares discriminant analysis (PLS-DA), and orthogonal projections to latent structures discriminant analysis (OPLS-DA) [14]. The differential metabolome between the two groups was analyzed based on the OPLS-DA score plot, and the differences in metabolites were screened according to the difference between the value of group contribution (VIP) and the significance (*p* < 0.05). The standard spectrum library of the National Institute of Standards and Technology (NIST) and the Wiley Registry metabolomic database [15] were used as reference.

#### 2.2.4. Flavoromics Analysis

Flavoromics analysis comprises a visual display of substances and their flavors after the flavor notes are determined for the substances identified in the two comparison groups. In the process of flavoromics annotation, the NIST database was used to identify substances in the offline data, and the Odor and FlavorDB databases were used to annotate the identified flavor substances [16], including their flavor names, the lowest and highest concentrations of flavor ranges, and sensory flavors. The sankey and radar diagrams were prepared using the ggplot package in R, and the network diagram was drawn using Cytoscape.

### 2.3. Contents of Numb Taste Components Analysis

#### 2.3.1. Samples Preparation

The fruits of the two *Zanthoxylum* samples were air-dried under laboratory conditions and then ground into small pieces. An amount of 100 g of the ground peels were extracted with 75% ethanol (3 × 200 mL) under reflux 3 times and filtered using a cotton plug followed by filter paper. *Z. bungeanum* fruit extract (22.3 g) and *Z. motuoense* fruit extract (15.7 g) were obtained by decompression and concentration. An amount of 100 mg of the extracts was weighed, dissolved in methanol, and diluted to 10 mL volumetric flasks. The sample solution to be tested was filtered with a 0.22 μm organic phase filter head, and 3 parallel groups were set for each sample.

#### 2.3.2. Preparation of Standard Solution

Hydroxy-α-sanshool, hydroxy-β-sanshool, and hydroxy-*γ*-sanshool were obtained from Yuanye Biotechnology Co., Ltd. Hydroxy-α-sanshool, hydroxy-β-sanshool, and hydroxy-*γ*-sanshool were dissolved in methanol to prepare a standard stock solution with a concentration of 1 mg/mL, and the stock the solutions were stored at −4 °C for standby.

The same amount of the above 3 standard stock solutions was suctioned and placed in the same volumetric flask and diluted with methanol to 100 μg/mL of mixed standard stock solution of 3 samples.

#### 2.3.3. Quantification of Hydroxy-α-sanshool, Hydroxy-β-sanshool, and Hydroxy-γ-sanshool via RP-HPLC

The contents of hydroxy-α-sanshool, hydroxy-β-sanshool, and hydroxy-*γ*-sanshool were quantified using Agilent Technologies 1260 series HPLC with a variable wavelength detector. The quantification was performed with a flow rate of 1.0 mL/min on a SB-C18 reversed-phase column (4.6 × 250 mm, 5 μm) at ambient temperature. The components of the mobile phase were methanol (solvent A), acetonitrile (solvent B), and phosphoric acid/water (1: 1000, *v*/*v*, solvent C). The first 50 min gradient from 55% solvent A, 2% solvent B to 43% solvent C, and the later 20 min gradient from 65% solvent A, 2% solvent B to 33% solvent C following a 20 min 100% solvent A were used to separate the different compounds. The injection volume was 10 μL at a wavelength of 270 nm. The standard curves of three sanshools were prepared by referring to the above chromatographic conditions. Three sanshools were identified by referring to the retention time and spectral characteristic peaks of the above chromatographic preparation conditions. The contents of hydroxy-α-sanshool, hydroxy-β-sanshool, and hydroxy-*γ*-sanshool showed good linear regression in the range from 20 to 1000 μg/mL: y = 3.917x + 30.392 (R2 = 0.9987), y = 4.4179x + 3.3323 (R2 = 0.9998), and y = 7.8504x + 19.582 (R2 = 0.9993) and expressed in terms of milligrams of hydroxy-α-sanshool, hydroxy-β-sanshool, and hydroxy-*γ*-sanshool per gram of samples powder, respectively (Appendix A). The precision of the method was calculated through testing the same sample six times. The repeatability was determined by analyzing six replicates of the sample [17].

### 2.4. Antimicrobial Activities

#### 2.4.1. Essential Oil Extraction

The essential oil extraction used for antimicrobial activity screening was performed with steam distillation [18]. Approximately 100 g of each sample was subjected to steam distillation for 4 h. The oils were isolated from the distillation water, dried with anhydrous Na_2_SO_4_, stored in a dark glass bottle, and kept at 4 °C until analysis. MEO essential oil (EO) was obtained from *Z. motuoense* pericarp with a yield of 2.5% (*v*/*w*), and BEO was obtained from *Z. bungeanum* pericarp with a yield of 3.0% (*v*/*w*).

#### 2.4.2. Tested Bacterial and Plant Pathogens

Bacterial strains: Staphylococcus aureus (ATCC 25923), *Escherichia coli* (ATCC 35218), Pseudomonas aeruginosa (ATCC 2753), and Ralstonia solanacearum (ATCC 11696).

Plant pathogen strains: *Sclerotinia sclerotiorum* (ATCC 18684), *Phytophthora parasitic* var. *Nicotiana* (ACCC 38065), *Fusarium graminearum* (ATCC 200362), *Cylindrocarpon destructans* (ATCC 36031), *Fusarium oxysporum* (ATCC 7601), *Cytospora mandshurica* (Bio-66226), *Phomopsis viticola* (Bio-21269), *Botryosphaeria dothidea* (Bio-19958), and *Alternaria tenuissima* (ATCC 51763). All tested strains were obtained from the Key Laboratory of Chemistry for Natural Products of Guizhou Province and Chinese Academy of Sciences (Guiyang, China).

#### 2.4.3. Antibacterial Activity Screening

All the strains were maintained on a Mueller–Hinton agar at 4 °C and were sub-cultured every month in a laboratory. The strains were cultured and examined via morphological and physiological characteristics experiments to ensure that the strains did not have any mutations before use. Bacterial strains were grown in modified Luria–Bertani (LB) medium (LB: 1% tryptone, 0.5% yeast extract, 0.5% NaCl; amended with 1.5% agar for solid media) for 1 day and dispersed in the sterile brine of the tube until the concentration was 0.5 times that of the McMalloy turbidimetric tube concentration [19,20].

The MIC values of EOs were determined via the broth microdilution method using 96-well microtiter plates according to the Laboratory Standards Institute guidelines [21,22]. Twofold serial dilutions of EOs were prepared in Mueller–Hinton (MH) broth (HiMedia). To each well, 5 μL of the inoculum (5 × 10^5^ CFU/mL) was added. Each plate contained growth and sterile control wells. The microtiter plates were incubated at 37 °C for 20 h and results were recorded. Ciprofloxacin was used as a positive control, and DMSO (1% *v*/*v*) was used as a negative control. The lowest concentration that completely inhibited growth, as detected by the unaided eye, was reported as the MIC. All assays were performed in duplicate.

#### 2.4.4. Antifungal Activity Screening

Antifungal activity was determined using the mycelial growth rate method, as described previously, with minor modifications [23]. Briefly, the fungi were sub-cultured on potato dextrose agar (PDA) three days before the experiment to prevent morphological and metabolic transformations. The EOs were diluted to final concentrations of 1.0, 0.8, 0.6, 0.4, 0.2, and 0.1 mg/mL in PDA supplemented with 1% agar, and the mixture was poured into 90 mm sterilized Petri dishes. Once solidified, 6 mm wells were cut from the medium with fungus and inoculated on the PDA culture medium. The agar plates were then incubated in the dark at 27 °C for 3 days, and the potential antifungal activity was evaluated. The diameters of the inhibition zones were measured when applicable. Each concentration was tested in triplicate to confirm the reproducibility of the results. The percentage of inhibition for each treatment was calculated from the average diameter of each repetition and compared with the average diameter of the cultures without treatment [24].

## 3. Results

HS-SPME/GC×GC-TOFMS was carried out to explore the volatile components of *Z. motuoense* pericarp. To elucidate the unique aroma of *Z. motuoense*, *Z. bungeanum* pericarp was used as a comparison. By using methods of multivariate statistical analysis and flavor-omics analysis, the differences in composition and flavor between *Z. motuoense* and *Z. bungeanum* were studied. The characteristic numb-taste components of two *Zanthoxylum* species were further elucidated with RP-HPLC. As a plant with important medicinal value, the medicinal use of *Z. bungeanum* in folk medicine is often related to antimicrobial activities. Therefore, we screened its antibacterial and antifungal activities subsequently.

### 3.1. Essential Oil Analysis by GC×GC-TOFMS

The optimized HS-SPME conditions were used to extract the EOs from the two Chinese prickly ashes, and the total ion flow chromatograms of MEO and BEO were obtained, as shown in Figure 2. There is a significant difference in the distribution of chemical components between the 2 samples, particularly in the area with a retention time of 20–35 min. The compositions of the MEO and BEO were analyzed using GC×GC-TOFMS and are listed in Appendix A. A total of 212 compounds were identified by comparing their retention indices and mass spectral information with those of the NIST Mass Spectral Library, including 56 alcohols, 35 terpenes, 34 esters, 33 aldehydes, 16 ketones, 7 acids, 8 alkanes, 8 amines, 3 alkynes, and 12 other compounds. The predominant components detected in the MEO were citronellal, (+)-citronellal, and β-phellandrene, followed by β-myrcene, citronellol acetate, limonene, D-limonene, isopulegol, allyl methyl ether, and (E,Z)-3,6-Nonadien-1-ol. The main components of the BEO were linalool, limonene, linalyl acetate, β-myrcene, 6-methyl-5-methylene-2-heptanone, γ-terpinene, β-phellandrene, 1-bromo-3-methyl-2-butene, trans-β-ocimene, and β-pinene (Appendix A). A similar result was reported for *Z. bungeanum* [5]. It can also be seen from Appendix A that the composition types and distributions of the two samples were similar, but the main components were significantly different, which might be the reason for their significant differences in aroma characteristics.

### 3.2. Multivariate Statistical Analysis

PCA is a multivariate statistics-based detection approach that uses the signal strengths of flavor substances to highlight differences between samples [25]. PCA analysis was performed to process the data of the two Chinese prickly ash samples to investigate the effect of the metabolites of the two species. In this experiment, the data were converted using Par (Pareto scaling) before multivariate statistical analysis to obtain more reliable and intuitive results. The computed PCA model, with no overfitting (Figure 3), allowed for a global and descriptive assessment of the distribution of samples to highlight natural groupings, trends, and outliers. As shown in Figure 3, the MEO sample was significantly distinct from the BEO sample. The first principal component (PC1) and second principal component (PC2) accounted for 66.1% and 14.0% of the variance, respectively. Together, these two components explained 80.1% of the total variance. All BEO samples were positioned on the right side and all MEO samples were positioned on the left side of the score plot.

For the biological characterization and interpretation of these informative differential metabolic profiles described by explorative modelling (PCA), an orthogonal partial least-squares discriminant analysis (OPLS-DA) method was used to establish a correlation model between the volatile compound content measured via HS-SPME-GC×GC-TOFMS and the sample category [26,27]. The OPLS-DA extends a regression of the PCA, uses the class membership to maximize the variation, and introduces an orthogonal signal correction (OSC) filter to separately handle the systematic variation correlated to, or uncorrelated to, the Y variable, Therefore, the OPLS-DA had better discriminant ability for the samples with larger within-class divergence than PCA [28]. In this experiment, as shown in Figure 4, the OPLS-DA model was built with high R2Y (0.996) and Q2 (0.977), indicating a good fit and high predictive ability and a low probability of model overfitting. As MEO was modeled separately from BEO, we could speculate that MEO was chemically different from BEO. The VIP value is generally used to explain the importance of variables in the model. When VIP > 1, the characteristic peak is important, which is usually considered as one of the screening conditions for potential biomarkers. According to the OPLS-DA, VIP > 1 and *p* < 0.05 were used as criteria to search for significantly differentially expressed metabolites [29].

In this experiment, the VIP value of the OPLS-DA model coupled with the *p*-value of the Student’s t-test was used to determine differential metabolites expression (Table 1). Simultaneously, a VIP (Variable Importance Projection) score higher than 1 was used as a screening criterion. Table 1 shows that 26 kinds of compounds were identified in total. Among these, linalool had the highest VIP value (5.24), followed by limonene (4.76), citronellal (4.46), linalyl acetate (4.23), and (R)-6-Octenal, 3,7-dimethyl-(4.16), which suggests they are the key potential aromatic compounds of these two species. The concentrations of six volatile compounds were significantly higher in MEO than in BEO. In contrast, the levels of twenty volatile compounds in MEO were significantly lower than those in BEO (Figure 5 and Figure 6). It is noteworthy that citronellal and (R)-6-Octenal, 3,7-dimethyl- were the characteristic compounds of MEO which even contributed considerably to the citrus-like aroma.

The above results further prove the differences in the main components between MEO and BEO. It also can be seen that citronellal, (E,Z)-3,6-nonadien-1-ol, allyl methallyl ether, isopulegol, 6-octen-1-ol, 3,7-dimethyl-, acetate, 6-octen-1-ol, and 3,7-dimethyl-(R)-6-octen-1-ol were the characteristic components of MEO and could be used as the potential biomarkers of MEO.

### 3.3. Flavoromics Profile and Sensory Analysis of the MEO and BEO

Plants can synthesize thousands of primary and secondary metabolites with diverse taste and smell properties. The main flavored metabolites in vegetables and fruits include sugars, acids, salts, bitter compounds, and volatiles. Among them, volatile compounds have gained increasing attention because of their crucial contribution to the unique flavors of vegetables and fruits. In contrast to taste-related chemicals, many volatile compounds are detected by humans at extremely low levels. Their chemical diversity, in combination with their large number of olfactory receptors, provides a variation in flavors that distinguishes individual foods in the human diet. A given fruit or vegetable can contain several different volatiles, and the unique combination within a given food determines its unique flavor profile [30,31]. Chinese prickly ash has occupied a very important position in the field of condiments due to its unique flavor. As a type of condiment with great developmental potential, it is necessary to study the differences in flavor between MEO and BEO using a flavoromics approach. Modified from chemometrics, flavoromics aims to clarify the key flavor compounds by collecting data, including, but not limited to, the chemical composition and sensory ratings of foods across a large group of samples. Multidimensional analytical platforms are then applied to integrate information and interpret the correlation between the collected observations. By combining instrumental analysis, sensory evaluation, and statistical modeling, flavoromics can screen chemical stimuli that are critical for flavor perception within a complex food matrix [32]. Flavor metabolomics is based on the GC-MS platform. GC-MS has high throughput, precision, sensitivity, and reproducibility [33]. It has a reference standard spectrogram database, which helps in a qualitative comparison. It can also detect most organic molecules in a sample. In this experiment, we employed a GC×GC/TOFMS full two-dimensional gas chromatography detection platform combined with the NIST database to identify the obtained data. The Odor and FlavorDB databases were used to annotate the identified flavor substances, including their flavor name, the lowest and highest concentrations of flavor range, and sensory flavor. The flavor substances identified in the two Zanthoxylum species are listed in Table 2. Eleven aromatic compounds were also detected. The odors of these compounds ranged from 0.00024 to 198686. Citronellal, D-limonene, isopulegol, linalool, and citronellol were the top five volatile compounds contributing to the flavor of MEO. Of these, citronellal and citronellol are monoterpenoids and are responsible for MEO’s distinctive lemony scent [34]. D-limonene has a weak, citrus-like aroma and is considered one of the major contributors to an orange flavor [35]. Isopulegol, a monoterpenic alcohol that presents aromas of camphor and mint, along with rose leaves and citronella notes, is widely used in the flavor industry for the production of fragrances in various pharmaceuticals. It has been established that isopulegol can be obtained from (+)-citronellal through cyclization [36,37]. Hence, the lemony, citrus-like, and mint aromas of the MEO were more prominent. In contrast, linalool, caryophyllene, terpinen-4-ol, D-limonene, and acetaldehyde were the top five volatile compounds contributing to the flavor of BEO. These compounds mainly provide the aroma of green, spicy, floral, lavender, orange, floral, woody, turpentine, pepper, musty, pungent, and mint and gave the samples a typical fruity odor. Furthermore, based on the correlations between the detected substances and sensory flavors, we established a Sankey map, a radar map, and a correlation network map (Figure 7 and Figure 8). The Sankey map (Figure 7) illustrated the key volatile components associated with key aroma types such as aldehydic, citrus, floral, fresh, etc. As can be seen from the radar map (Figure 8), compared with BEO, MEO had significantly higher floral, fresh, and herbal notes, especially in terms of a fresh aroma, which did not exist in BEO. In contrast, BEO had predominantly woody, waxy, fruity, citrus, and green notes.

The correlation network map (Figure 8) illustrates the relationship between the key aroma components, odor categories, and flavors in different groups. Compared with BEO, citronellal, isopulegol, and citronellol showed very high contents in MEO and contributed significantly to its aroma. These components may be the main influencing factors that affect the difference between the odors of MEO and BEO. Despite the differences in major components, both species have three common compounds with the highest impact on aroma: β-phellandrene, β-myrcene, and limonene.

### 3.4. Quantification of Hydroxy-α-sanshool, Hydroxy-β-sanshool, and Hydroxy-γ-sanshool in the Z. motuoense and Z. bungeanum Peel

Numerous studies have shown that the numb taste of *Z. bungeanum* is evoked by aliphatic alkylamides, which were constructed of unsaturated fatty acid chains and N-terminal isobutyl structures [38]. Of these, hydroxy-α-sanshool, hydroxy-β-sanshool, and hydroxy-γ-sanshool are the alkylamides forming the main numb-taste substances in *Z. bungeanum*. In order to further study the numb taste substances that affect the flavor of *Z. motuoense*, the contents of hydroxy-α-sanshool, hydroxy-β-sanshool, and hydroxy-γ-sanshool in the peel from the two *Zanthoxylum* species were determined via RP-HPLC (see Table 3, Figure 9 and Appendix A). The results show that the contents of hydroxy-α-sanshool, hydroxy-β-sanshool, and hydroxy-γ-sanshool in *Z. motuoense* were 1.31 ± 0.07, 2.61 ± 0.16, and 77.91 ± 5.79 mg/g, respectively. The contents of hydroxy-α-sanshool, hydroxy-β-sanshool, and hydroxy-γ-sanshool in *Z. bungeanum* were 78.71 ± 5.59, 151.23 ± 11.67, and 1.79 ± 0.12 mg/g, respectively. These results suggested that the contents of hydroxy-α-sanshool and hydroxy-β-sanshool in *Z. motuoense* were well below those of *Z. bungeanum*. However, the content of hydroxy-γ-sanshool was much higher than that of *Z. bungeanum*. The total sanshool contents in *Z. motuoense* and *Z. bungeanum* were calculated as 81.83 ± 6.02 and 231.73 ± 17.38 mg/g, respectively. This result may well explain why the pungent taste of *Z. motuoense* is much less pronounced than that of *Z. bungeanum*. This lightly pungent taste can meet the needs of consumers who are more sensitive to taste.

### 3.5. Antibacterial Activity

Previous research has reported that BEO shows a variable degree of antibacterial activities against *S. aureus*, B. laubach (MIC = 1.25 mg/mL), *B. subtilis*, *B. cereus*, and *E. coli* with an MIC range of 1.25~2.5 mg/mL [15]. In this study, we screened the antibacterial activities of MEO and BEO against four bacterial strains. The results (Table 4) showed that MEO had higher inhibitory activity against three bacterial strains (*S. aureus*, *E. coli*, and *R. solanacearum*) than BEO, with MIC values of 0.375, 0.188, and 0.750 mg/mL, respectively, and with a similar MIC value of the two samples against *P. aeruginosa* (1.5 mg/mL).

### 3.6. Antifungal Activity

For many years, synthetic fungicides have been used for the control of plant pathogenic fungi. However, the extensive use of these chemicals has led to the development of resistance and environmental damage in many areas around the world [39]. Decreasing efficacy and increasing concern over the adverse environmental effects of synthetic fungicides have brought about the need for the development of new types of selective control alternatives and crop protection methods with or without the reduced use of conventional fungicides. Essential-oil-bearing plants may constitute an alternative to currently used disease control agents since they constitute a rich source of bio-active chemicals [40,41].

The antifungal activities of MEO and BEO were tested against nine phytopathogenic fungal strains: *S. sclerotiorum*, *P. parasitic* var. *Nicotianae*, *F. graminearum*, *F. oxysporum*, *C. destructans*, *C. mandshurica*, *P. viticola*, *B. dothidea*, and *A. tenuissima*. The samples were added to each Petri dish, which allowed only volatiles to be the causative agents for any microbial inhibition. The preliminary screening results of the two essential oils against the nine fungal strains at a concentration of 1 mg/mL are shown in Table 5. The results showed that MEO can completely inhibit the mycelial growth of *S. sclerotiorum*, *P. parasitic* var. *Nicotianae*, *F. graminearum*, and *C. destructans* at 1 mg/mL, and the inhibition rates on *F. oxysporum* and *C. mandshurica* were 68.19% and 65.37%, respectively. Furthermore, the inhibition rate of MEO on *P. viticola*, *B. dothidea*, and *A. tenuissima* was less than 50% at 1 mg/mL. In contrast, the inhibition rate of mycelial growth on *P. parasitic* var. *Nicotianae*, *F. graminearum*, *C. destructans*, *F. oxysporum*, and *C. mandshurica* of BEO were significantly lower than those of MEO at 1 mg/mL, with fungal growth inhibitory activities ranging from 21.76% to 81.54%. Based on the preliminary screening results, the concentration-dependent effects of the two samples on the mycelial growth of the seven fungi were evaluated. The results (Table 6) showed that MEO had a significant inhibitory effect on the mycelial growth of S. sclerotiorum, *F. graminearum*, and *P. parasitic* var. *Nicotianae*, with the IC_50_ values of 0.22 ± 0.04 mg/mL, 0.23 ± 0.03 mg/mL, and 0.27 ± 0.02 mg/mL, respectively. The inhibitory effect of MEO on the mycelial growth of *C. destructans*, *F. oxysporum*, and *C. mandshurica* corresponded to IC_50_ values of 0.39 ± 0.02, 0.51 ± 0.06, and 0.72 ± 0.09 mg/mL, respectively. However, the inhibitory activity of BEO against these fungal strains was much lower than that of MEO.

*Z. motuoense* essential oil showed a noticeable effect against the tested microorganisms, demonstrating the possibility of using these samples as antimicrobial agents when combined with commercial antibiotics for crop disease defense.

## 4. Conclusions

In this study, the volatile components of *Z. motuoense* were first characterized via HS-SPME/GC×GC-TOFMS coupled with multivariate data analysis, including PCA, PLS-DA, and OPLS-DA, and a flavoromics approach. The common commercial Chinese pepper *Z. bungeanum* was used for comparison. MEO was mainly shown to be composed of alcohols, terpenoids, esters, aldehydes, and ketones. Comparing the two Chinese prickly ashes twenty differential metabolites were identified: six unique metabolites were found in MEO, which can be used as the biomarkers of *Z. motuoense*. Citronellal was the most abundant aroma component in MEO. The flavoromics approach further verified the difference between the two sensory aroma types and helped analyze the relationship between the main volatile components as well as different aroma types. MEO had significantly higher floral, fresh, and herbal notes, especially in terms of a fresh aroma, which did not exist in BEO. The quantification of the three main pungency sensation substances, hydroxy-α-sanshool, hydroxy-β-sanshool, and hydroxy-γ-sanshool, suggested that the contents of hydroxy-α-sanshool and hydroxy-β-sanshool of *Z. motuoense* were well below those of *Z. bungeanum*. This may well explain why the pungent taste of *Z. motuoense* is much less pronounced than that of *Z. bungeanum*. The in vitro antibacterial and anti-fungal activities of MEO were assessed for the first time against four bacterial strains and nine plant pathogenic fungi strains in crops. The results show that MEO had a significantly higher inhibitory activity against most fungi strains than BEO did. Based on the above results, it can be seen that the aroma of *Z. motuoense* is significantly different from that of *Z. bungeanum*. It is a seasoning plant with high economic value and has high potential in the development and exploration of medicinal effects, especially in the research of inhibiting agricultural pathogenic fungi. It is necessary to continue to conduct in-depth research on its in vivo antifungal activity screening and the mechanism research, especially with respect to some challenging agricultural pathogenic fungi.

## Figures and Tables

**Figure 1 foods-12-02225-f001:**
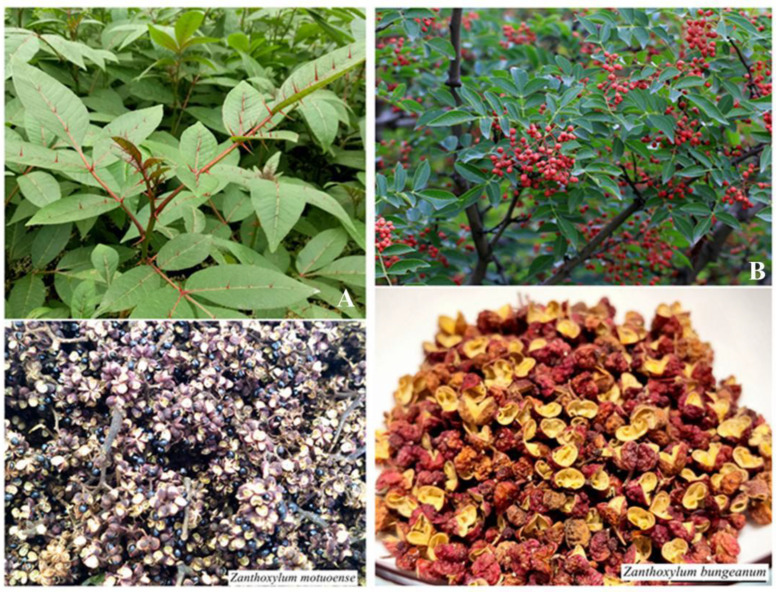
Morphological images of *Z. motuoense* and *Z. bungeanum*. (**A**) left: *Z. motuoense*; (**B**) right: *Z. bungeanum*.

**Figure 2 foods-12-02225-f002:**
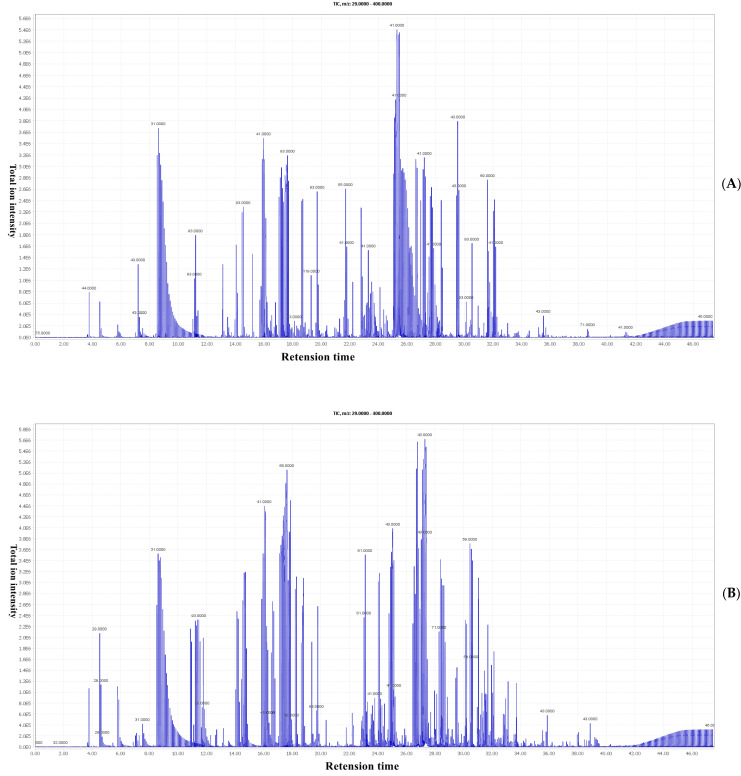
The GC-MS total ion chromatogram of the two samples. (**A**): MEO; (**B**): BEO.

**Figure 3 foods-12-02225-f003:**
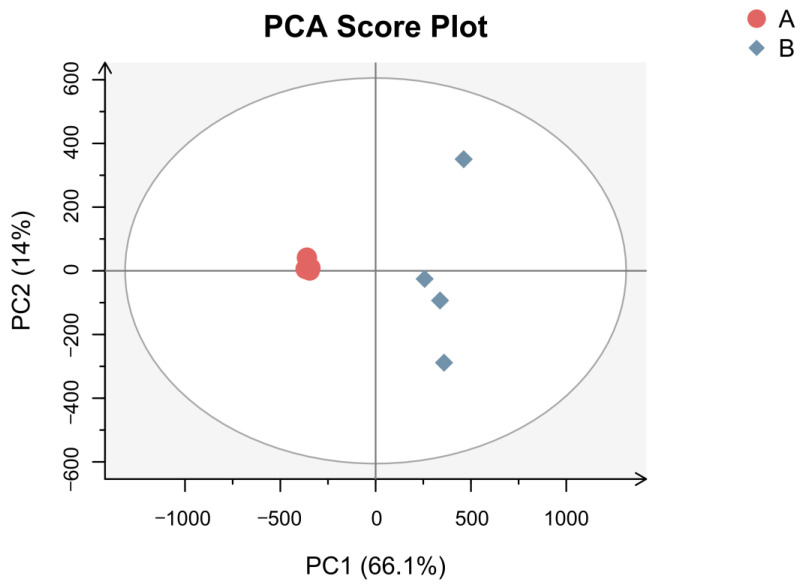
The PCA score plot of MEO and BEO. A: MEO; B: BEO.

**Figure 4 foods-12-02225-f004:**
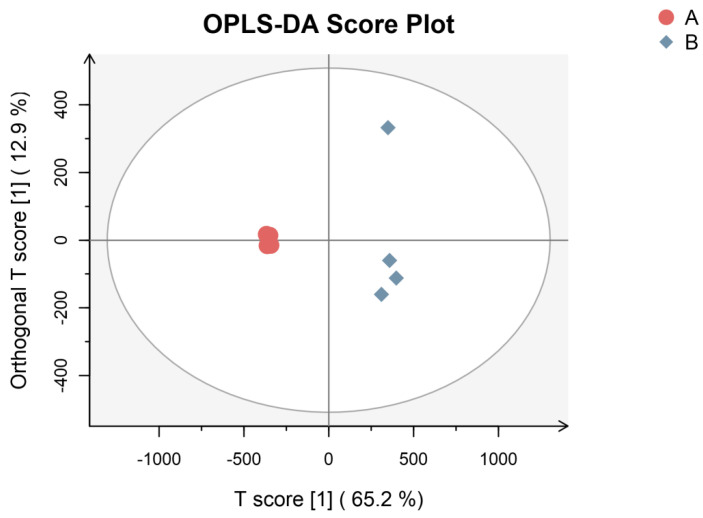
OPLS-DA Score plot of MEO (A) and BEO (B) (R^2^X = 0.78, R^2^Y = 0.996, Q^2^ = 0.977).

**Figure 5 foods-12-02225-f005:**
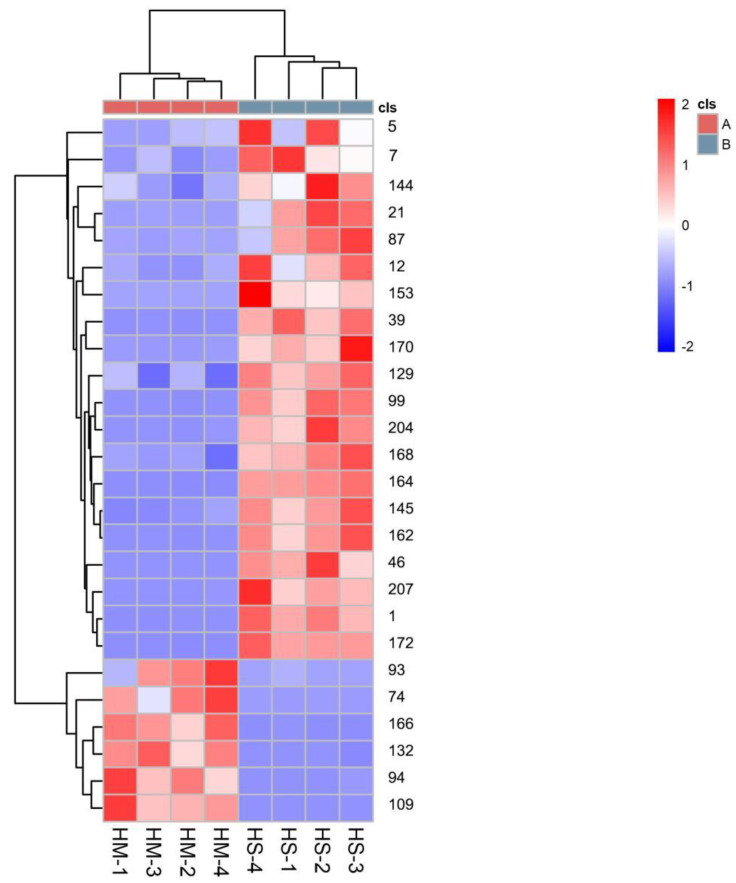
A heat map for the key compounds (VIP > 1 and *p* < 0.05) and HCA for the MEO (A) and BEO (B). The number represents the identified metabolites in Appendix A. The red color indicates a relatively high content. The blue color indicates low content of metabolites.

**Figure 6 foods-12-02225-f006:**
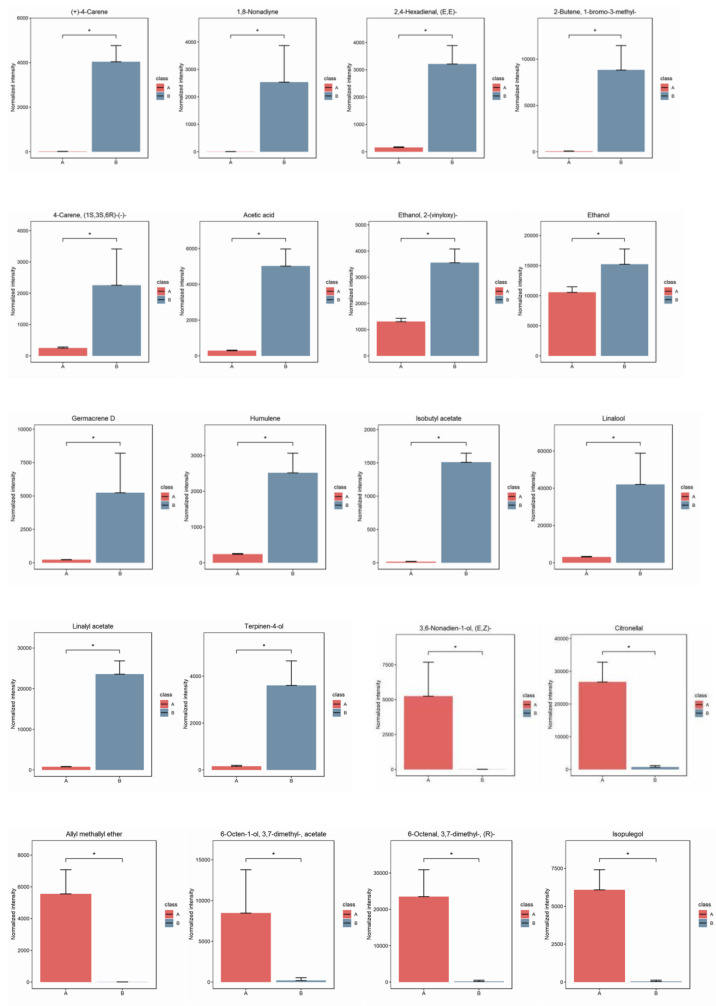
Metabolites with significant differences between MEO (A) and BEO (B). Red indicates metabolites in MEO, and blue indicates metabolites in BEO. The compounds notes with “*” are significantly different between samples A and B (*p* < 0.05).

**Figure 7 foods-12-02225-f007:**
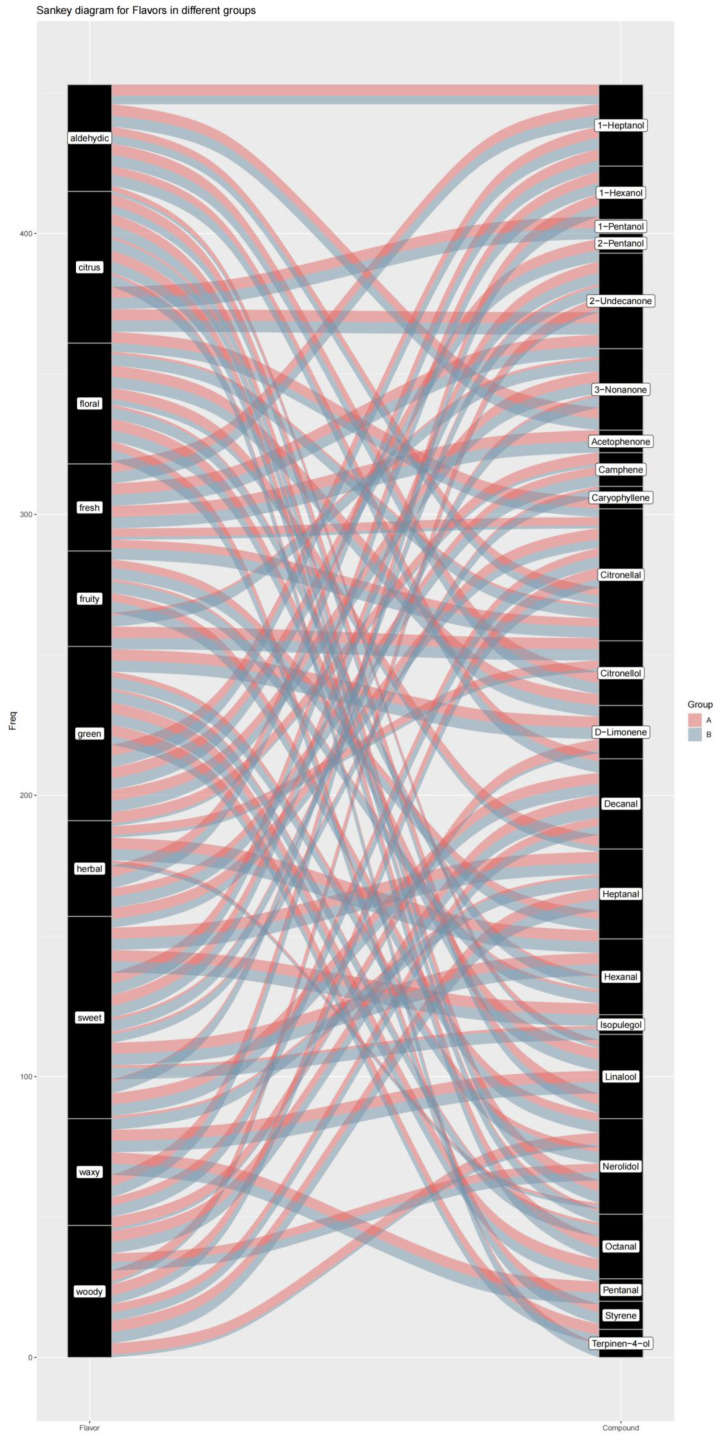
The correlations between detection substances and sensory flavor in MEO and BEO ((A): MEO, (B): BEO).

**Figure 8 foods-12-02225-f008:**
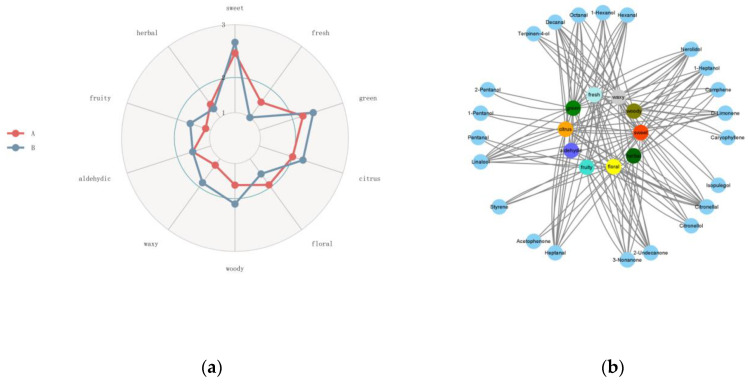
(**a**) The aroma profiles of “Huajiang” samples obtained from MEO and BEO (A: MEO, B: BEO); (**b**) The relationship between the main volatile components of two kinds of Zanthoxylum species and different aroma types. Blue circle: volatiles. The other color circles: aroma types.

**Figure 9 foods-12-02225-f009:**
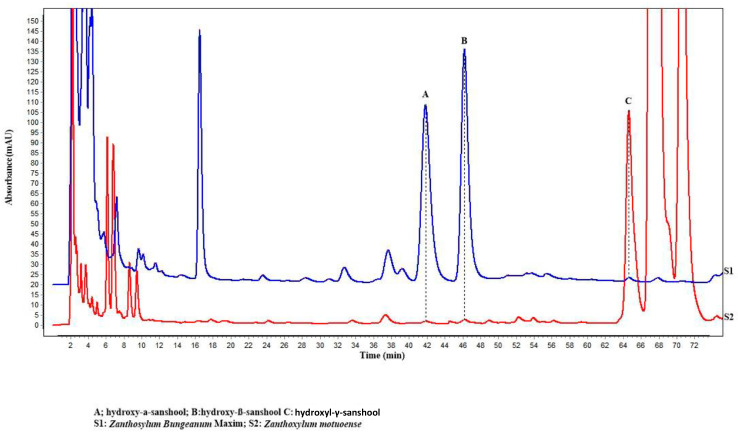
HPLC-fingerprint of two species of *Zanthoxylum* fruits (Peaks A, B, and C were identified as hydroxyl-α-sanshool, hydroxyl-β-sanshool, and hydroxyl-γ-sanshool, respectively).

**Table 1 foods-12-02225-t001:** Differential metabolites of MEO and BEO.

NO.	Compounds	CAS	rt (min)	Molecular Weight	Molecular Formula	VIP	*p*-Value	log2(FC_A/B)
1	(+)-4-Carene	29050-33-7	28.5	136.23	C_10_H_16_	1.770693031	0.03	−8.6689
2	*α*-Pinene	80-56-8	19.6667	136.23	C_10_H_16_	1.540234605	0.029	−3.443
3	*β*-Myrcene	123-35-3	16	136.23	C_10_H_16_	3.131640524	0.03	−1.3939
4	*γ*-Terpinene	99-85-4	25.0833	136.23	C_10_H_16_	2.456300509	0.03	−3.1802
5	1,8-Nonadiyne	2396-65-8	19.25	120.19	C_9_H_12_	1.302772377	0.029	−9.1911
6	(E,E)-2,4-Hexadienal	142-83-6	23	96.13	C_6_H_8_O	1.532866599	0.03	−4.3574
7	2-Butene, 1-bromo-3-methyl-	870-63-3	20.25	149.03	C_5_H_9_Br	2.560639238	0.03	−7.4076
8	(E,Z)-3,6-Nonadien-1-ol	56805-23-3	25.4167	140.23	C_9_H_16_O	1.905607225	0.029	+9.24
9	4-Carene, (1S,3S,6R)-(−)-	5208-50-4	25	136.23	C_10_H_16_	1.143406224	0.03	−3.1904
10	6-Octen-1-ol, 3,7-dimethyl-, acetate	150-84-5	29.4167	198.30	C_12_H_22_O_2_	2.277011945	0.026	+5.5667
11	(R)-6-Octenal, 3,7-dimethyl-	2385-77-5	25.3333	154.25	C_10_H_18_O	4.163651501	0.026	+7.0734
12	Acetic acid	64-19-7	24.0833	60.05	C_2_H_4_O_2_	1.913005062	0.03	−4.118
13	Allyl methallyl ether	14289-96-4	25.4167	112.17	C_7_H_12_O	2.048043888	0.03	+9.961
14	Caryophyllene	87-44-5	28.3333	204.36	C_15_H_24_	1.661784003	0.029	−2.7406
15	Citronellal	106-23-0	25.3333	154.25	C_10_H_18_O	4.464295154	0.03	+5.2502
16	Ethanol	64-17-5	8.58333	46.07	C_2_H_6_O	1.741290164	0.03	−0.52856
17	Ethanol, 2-(vinyloxy)-	764-48-7	47.1667	88.11	C_4_H_8_O_2_	1.310679373	0.03	−1.4438
18	Germacrene D	23986-74-5	30.9167	204.35	C_15_H_24_	1.799569472	0.03	−4.5682
19	Humulene	6753-98-6	30.0833	204.36	C_15_H_24_	1.319712086	0.03	−3.4072
20	Isobutyl acetate	110-19-0	10.8333	116.16	C_6_H_12_O_2_	1.087154338	0.03	−6.5814
21	Isopulegol	89-79-2	27.75	154.25	C_10_H_18_O	2.157177505	0.026	+7.152
22	Limonene	138-86-3	17.0833	136.23	C_10_H_16_	4.755266283	0.03	−2.8155
23	Linalool	78-70-6	26.5833	154.25	C_10_H_18_O	5.238708648	0.03	−3.7743
24	Linalyl acetate	115-95-7	26.9167	196.28	C_10_H_20_O	4.231364101	0.03	−4.9328
25	Terpinen-4-ol	562-74-3	28.25	154.25	C_10_H_18_O	1.606004596	0.03	−4.5477
26	*trans*-*β*-Ocimene	3779-61-1	11.25	136.23	C_10_H_16_	2.117693803	0.026	−9.3875

rt: retention time; VIP: the contribution rate of different substances to the OPLS-DA model; *p*-value: a significance value for the *t*-test; log2(FC_A/B): log2 of the ratio of the mean value of the MEO to the construction of BEO; +: rising; −: falling.

**Table 2 foods-12-02225-t002:** The flavor substances identified from MEO and BEO.

Name	CAS	rt (min)	Range of Odor Min	Range of Odor Max	Odor Character	Flavor_Profile	MEO	BEO
Citronellal	106-23-0	25.3333				herbal, dry, fat, rose, aldehydic, floral, sweet, cherry, lemon, citrus, waxy	26,709.40	935.64
D-Limonene	5989-27-5	25.3333				mint, fresh, orange, sweet, lemon, citrus	6174.66	3558.32
Isopulegol	89-79-2	27.75				medicinal, cooling, minty, woody	6089.05	0.15
Linalool	78-70-6	26.5833				green, flower, lavender, orange, floral, sweet, lemon, blueberry, citrus, bois de rose, woody	3070.13	42,008.66
Citronellol	106-22-9	32				leather, rose, bitter, floral, citrus, rose bud, geranium, waxy	2583.72	0.15
Decanal	112-31-2	25.6667				tallow, aldehydic, floral, sweet, citrus, soap, orange peel, waxy	1468.81	218.80
Caryophyllene	87-44-5	28.3333				spicy, wood, woody	1289.66	4310.19
Acetaldehyde	75-07-0	47.1667	0.0015	1000	pungent, fruity, suffocating, fresh, green		954.74	1437.09
1-Heptanol	111-70-6	26.0833				herbal, violet, green, leafy, coconut, sweet, peony, musty, chemical, strawberry, woody	363.65	149.53
Octanal	124-13-0	19.75				fat, green, aldehydic, lemon, citrus, fatty, soap, orange peel, waxy	312.65	76.40
Formaldehyde	50-00-0	47.0833	0.027	9770	pungent		207.04	560.89
Terpinen-4-ol	562-74-3	28.25				turpentine, pepper, must, sweet, nutmeg, musty, earth, woody	154.19	3606.38
Hexanal	66-25-1	13.0833				leafy, fruity, sweaty, grass, fatty, aldehydic, green, fresh, tallow, fat	135.52	137.42
Propylene Glycol	57-55-6	40.5833	5.14	0			128.37	123.91
Methyl Alcohol	67-56-1	7.5	3.05	198,686	sour, sweet, alcohol		95.71	344.33
Benzaldehyde	100-52-7	26.25	0.0015	783	bitter almond, fruit, vanilla		54.01	51.20
Propane	74-98-6	43.0833	1497	19,964	natural gas		47.37	22.32
1-Octanol	111-87-5	27.25	0.0009	1.69	penetrating		42.06	34.52
Heptanal	111-71-7	16.5				citrus, wine-lee, rancid, ozone, fatty, herbal, aldehydic, green, fresh, fat	27.70	11.11
1-Hexanol	111-27-3	21.75				fruity, ethereal, green, flower, sweet, alcoholic, resin, fusel, oil	23.22	32.07
Acetonitrile	75-05-8	10.4167	13	1161	etherish		21.08	42.46
Camphene	79-92-5	12.5833				herbal, camphor, fir needle, woody	12.20	383.64
Pentanal	110-62-3	9.75				fruity, malt, bready, nutty, pungent, almond, fermented, berry	11.59	30.33
Nerolidol	142-50-7	37.4167				green, wax, flower, floral, citrus, wood, woody, waxy	7.34	0.15
2-Undecanone	112-12-9	32.8333				fruity, green, creamy, fresh, orange, orris, floral, fatty, iris, waxy	6.57	0.15
Isoprene	78-79-5	23.9167	0.047	3.59	aromatic		5.25	6.48
1-Pentanol	71-41-0	18.6667				balsamic, vanilla, balsam, sweet, fusel, oil	4.34	12.25
2-Pentanol	6032-29-7	32.3333				green, mild, fusel, fermented, oil	2.80	3.90
Styrene	100-42-5	18.8333				balsamic, floral, balsam, sweet, plastic, gasoline	1.49	0.15
Furfural	98-01-1	23.5833	0.002	0.713	bread, almond		1.16	41.56
Acetophenone	98-86-2	29.4167	0.00024	0.59	sweet, almond, pungent, oranges, river water	acacia, flower, must, pungent, hawthorn, almond, bitter, sweet, hawthorne, mimosa, chemical	0.98	144.54
3-Nonanone	925-78-0	21.75				herbal, leaf, fruity, fresh, sweet, jasmin, spicy	0.15	87.48

**Table 3 foods-12-02225-t003:** Analysis results of sanshools contents in two species of *Zanthoxylum* fruits.

No.	Samples	Hydroxy-α-sanshool (mg/g)	Hydroxy-β-sanshool (mg/g)	Hydroxy-γ-sanshool (mg/g)	Total Sanshools Content (mg/g)
1	*Zanthoxylum motuoense*	1.31 ± 0.07	2.61 ± 0.16	77.91 ± 5.79	81.83 ± 6.02
2	*Zanthosylum bungeanum*	78.71 ± 5.59	151.23 ± 11.67	1.79 ± 0.12	231.73 ± 17.38

**Table 4 foods-12-02225-t004:** Evaluation of antibacterial activity in vitro.

Bacterial	MIC (mg/mL)	
	MEO	BEO
*Staphylococcus aureus*	0.375	0.750
*Pseudomonas aeruginosa*	1.500	1.500
*Escherichia coil*	0.188	0.750
*Ralstonia solanacearum*	0.750	1.500

MIC: minimal inhibitory concentration.

**Table 5 foods-12-02225-t005:** In vitro antifungal activity evaluation of MEO, BEO at 1 mg/mL.

Fungus	Inhibition Rate (%)		
	MEO	BEO	Carbendazim ^1^
*Sclerotinia sclerotiorum*	100.00	100.00	100.00
*Phytophthora parasitic var. Nicotianae*	100.00	74.36	100.00
*Fusarium graminearum*	100.00	81.54	100.00
*Cylindrocarpon destructans*	100.00	27.16	100.00
*Fusarium oxysporum*	68.19	57.26	100.00
*Cytospora mandshurica*	65.37	55.88	100.00
*Phomopsis Viticola*	<50	55.33	100.00
*Botryosphaeria dothidea*	<50	<50	86.81
*Alternaria tenuissima*	<50	57.48	100.00

^1^ Carbendazim was used as a positive control.

**Table 6 foods-12-02225-t006:** IC_50_ values of MEO and BEO on the mycelial growth rate of the plant pathogens.

Fungus	IC_50_ (μg/mL)		
	MEO	BEO	Hymexazol *
*Sclerotinia sclerotiorum*	220 ± 40 ^c,E^	536 ± 9 ^a,E^	49 ± 2.5
*Phytophthora parasitic* var. *Nicotianae*	270 ± 18 ^b,D^	1000 ± 158 ^a,B^	7 ± 0.9
*Fusarium graminearum*	230 ± 28 ^c,E^	780 ± 21 ^a,D^	25 ± 3.8
*Cylindrocarpon destructans*	390 ± 17 ^b,C^	-	37 ± 1.9
*Fusarium oxysporum*	511 ± 59 ^c,B^	938 ± 59 ^a,C^	28 ± 4.8
*Cytospora mandshurica*	718 ± 88 ^b,A^	1129 ± 69 ^a,A^	21 ± 1.6
*Alternaria tenuissima*	-	951 ± 70 ^a,C^	45 ± 1.8

- no active, * Hymexazol was used as positive control; control: blank PDA medium; (a–c) different lowercase letters indicate significant differences between samples in fungal IC_50_ (*p* < 0.05); (A–E) different uppercase letters indicate significant differences for the same sample within different fungal IC_50_ (*p* < 0.05).

## Data Availability

Data is contained within the article or Appendix A.

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
