# Peer review of "HS-SPME/GC×GC-TOFMS-Based Flavoromics and Antimicrobial Properties of the Aroma Components of Zanthoxylum motuoense"

_foods, 2023, doi:10.3390/foods12112225_

Round 1

Reviewer 1 Report

The work includes the study of the volatile compounds (essential oils) of the pericarp of Z. motuoense (MEO) and the commercial Chinese prickly ash in Asia Zanthoxylum bungeanum (BEO) analysed by HS-SPME /GC×GC-TOFMS in conjunction with multivariate data and flavoromics analyses. The antimicrobial activity of MEO and BEO against various strains of bacteria and strains of agricultural pathogenic fungi was also tested.

The work lacks the complete composition of the essential oils obtained, with the percentages of each ingredient given. Considering all the techniques used and available in the research, I consider this a major shortcoming. The composition of the volatile components affects their properties, and this is a very important part of these studies that is missing here. Experimental studies have also been done, and they are not well explained, but more emphasis is placed on statistical analysis. For a high quality work, it is necessary to first explain the experimental results obtained and then compare them with the statistically obtained values, which contributes to the quality of the research. To improve the work, it is necessary to pay attention to these corrections, i.e. to explain more thoroughly the obtained experimental results.

In addition, it is necessary to write chemical compounds according to IUPAC recommendations, and it is also necessary to check and correct them if necessary.

As I state above, it is necessary to improve the English language a little

Author Response

Thank you for your kindly comments and suggestions regarding our manuscript. We have modified the manuscript accordingly, and the amendments were marked up using the “Track Changes” function. The detailed corrections are listed below point by point:

Reviewer: 1

Comments to the Author

The work lacks the complete composition of the essential oils obtained, with the percentages of each ingredient given. Considering all the techniques used and available in the research, I consider this a major shortcoming. The composition of the volatile components affects their properties, and this is a very important part of these studies that is missing here. Experimental studies have also been done, and they are not well explained, but more emphasis is placed on statistical analysis. For a high quality work, it is necessary to first explain the experimental results obtained and then compare them with the statistically obtained values, which contributes to the quality of the research. To improve the work, it is necessary to pay attention to these corrections, i.e. to explain more thoroughly the obtained experimental results.

In addition, it is necessary to write chemical compounds according to IUPAC recommendations, and it is also necessary to check and correct them if necessary.

Response:  Thanks for your kindly comments.

The complete composition and the percentages of each ingredient of the essential oils obtained have been added to Table S1 of the supplementary materials. In addition, the experimental results part has been added a more comprehensive explanation accordingly. The chemical compounds have been checked according to IUPAC carefully.

Thanks for your consideration, and please contact me if you have any questions.

Kind regards,

Wei Gu, PhD

Reviewer 2 Report

Please see attached pdf file with comments.

Kind regards.

Author Response

Dear reviewer:

Thank you very much for the suggestions marked in the manuscript. We have carefully checked and revised accordingly.

Thanks for your consideration, and please contact me if you have any questions.

Kind regards,

Wei Gu, PhD

Reviewer 3 Report

The article is interesting; however the multivariate analysis was carried out by analyzing a technical quadruplicate of the each sample; therefore, no real biological replicates was included. This is a major flaw in the study design, affecting the quality of all results and diminishing the scientific soundness of the paper.

The English writing must be improved throughout the manuscript.

Author Response

Dear reviewer:

Thank you for your kindly comments and suggestions regarding our manuscript. We have modified the manuscript accordingly, and the amendments were marked up using the “Track Changes” function. The detailed corrections are listed below point by point:

The article is interesting; however the multivariate analysis was carried out by analyzing a technical quadruplicate of the each sample; therefore, no real biological replicates was included. This is a major flaw in the study design, affecting the quality of all results and diminishing the scientific soundness of the paper.

The English writing must be improved throughout the manuscript.

Response:  Thanks for your kindly comments.

We have conducted 4 biological replicates on each group of samples, and the relevant data has been added and improved in Table S1 of the supplementary materials. Please check and review.

In addition, the English writing have be improved according to the editing services.

Thanks for your consideration, and please contact me if you have any questions.

Kind regards,

Wei Gu, PhD

Round 2

Reviewer 1 Report

Dear Sir,

First of all, I emphasize the need to highlight the changed parts of the work. In other words, it should be clearly emphasized which part of the work was changed and in what way.

Moreover, my comment referred to the discussion, not to the experimental part. Table S1 is extensive and it is necessary to highlight the most important parts from it in the results.

This work is entirely devoted to statistical data processing and is less based on experimental data. Therefore, I still think that the work should be refined, i.e., the discussion related to the experimentally obtained data should be extended.

After these corrections, the work can be accepted for publication.

Author Response

Thank you for your kindly comments and suggestions regarding our manuscript. We have modified the manuscript accordingly, and the amendments were marked up using the “Track Changes” function. The detailed corrections are listed below point by point:

Comments to the Author

1 First of all, I emphasize the need to highlight the changed parts of the work. In other words, it should be clearly emphasized which part of the work was changed and in what way.

Response:  Thanks for your kindly comment.

The PDF document With modification marks has been uploaded as required.

2 Moreover, my comment referred to the discussion, not to the experimental part. Table S1 is extensive and it is necessary to highlight the most important parts from it in the results.

Response:  Thanks for your kindly comment.

The most important parts of the results in Table S1 have been added to Supplementary Materials as Table S2 and Table S3, respectively, and have been added in the sections 3.1, 3.3, 3.4 and 3.6 of the text.

3 This work is entirely devoted to statistical data processing and is less based on experimental data. Therefore, I still think that the work should be refined, i.e., the discussion related to the experimentally obtained data should be extended.

Response:  Thanks for your kindly comment.

The discussion of the experimental results has been extended in the corresponding sections. Please refer to the modified marking section for details.

Thanks for your consideration, and please contact me if you have any questions.

Kind regards,

Wei Gu, PhD

Reviewer 3 Report

The authors included all recommendations suggested by the reviewers.

No.

Author Response

Response to  reviewer 2 comment

Thank you very much for your recognition.

Wei Gu